# Extension of the Complete Data Fusion algorithm to tomographic retrieval products

Cecilia Tirelli<sup>1</sup>, Simone Ceccherini<sup>1</sup>, Samuele Del Bianco<sup>1</sup>, Bernd Funke<sup>2</sup>, Michael Höpfner<sup>3</sup>, Ugo Cortesi<sup>1</sup>, and Piera Raspollini<sup>1</sup>

Correspondence: Cecilia Tirelli (c.tirelli@ifac.cnr.it), Simone Ceccherini (s.ceccherini@ifac.cnr.it)

#### Abstract.

In data analysis of atmospheric remote sensing, the combination of complementary measurements of the same atmospheric state from different sensors operating with different geometries and/or in different spectral ranges is a powerful technique to advance the knowledge of tropospheric and stratospheric processes. The Complete Data Fusion (CDF) is an a posteriori method used so far to combine only one-dimensional atmospheric products (vertical profiles) related to the same or nearby geolocations from simultaneous and independent remote sensing observations. In this study, we demonstrate the applicability of the CDF algorithm to two-dimensional products and show its first application to simulated ozone datasets from the future Infrared Atmospheric Sounding Interferometer New Generation (IASI-NG) mission and the Changing-Atmosphere Infrared Tomography (CAIRT) ESA's Earth Explorer 11 candidate mission, in nadir- and limb-viewing observational geometry, respectively. We present the analysis of the performance of the CDF in three (one one-dimensional and two two-dimensional) case studies considering different configurations for the acquisitions of the two sensors, evaluating for each the number of degrees of freedom, the Shannon information content, the total errors and the spatial resolution. Furthermore, we quantitatively compare the 1D-CDF and the 2D-CDF performances, demonstrating that the exploitation of tomographic capabilities of atmospheric sensors allows advanced data fusion techniques, like 2D-CDF, to maximize the information extracted from complementary datasets.

## 1 Introduction

In recent years, the availability of a huge amount of data from remote sensing missions strengthened the continuous monitoring of the atmosphere and stimulated the use of new methods to gain the largest amount of information from these global measurements and to reduce large volumes of data. One of the possible approaches is to combine remote sensing measurements of the same air mass in order to obtain a single product from several measurements (Aires 2011, Aires et al. 2012). The benefits of combining complementary measurements, due to their different observation geometries (nadir and limb) and spectral ranges

<sup>&</sup>lt;sup>1</sup>Istituto di Fisica Applicata "Nello Carrara" del Consiglio Nazionale delle Ricerche, Via Madonna del Piano 10, 50019 Sesto Fiorentino, Italy

<sup>&</sup>lt;sup>2</sup> Instituto de Astrofísica de Andalucía, CSIC, Spain

<sup>&</sup>lt;sup>3</sup>Institute of Meteorology and Climate Research, Karlsruhe Institute of Technology, Postfach 3640, 76021 Karlsruhe, Germany

(from far IR to UV), have been demonstrated in recent studies using both real and simulated data (Staelin and Kerekes 1995, Warner et al. 2014, Cuesta et al. 2013, Worden et al. 2007, Costantino et al. 2017, Cuesta et al. 2018, Cuesta et al. 2022, Okamoto et al. 2023, Mettig et al. 2022, Zhao et al. 2022, Liu et al. 2022, Okamoto et al. 2023, Hache et al. 2014, Natraj et al. 2011, Landgraf and Hasekamp 2007, Sato et al. 2018, Fu et al. 2013, Fu et al. 2016, Cortesi et al. 2016, Sofieva et al. 2022). Two classes of strategies are widely used for the data combination: the synergistic retrieval and the a posteriori combination of the retrieved products. The synergistic retrieval (i.e., the simultaneous retrieval of all the measurements that are combined), even though it rigorously combines the complementary information of the measurements, suffers from a complex and costly implementation. It requires to integrate into a single inversion system the radiative transfer models capable to simulate the measurements of all the sensors involved in the synergistic inversion, implying the need to handle relevant Level 1 data volumes. The advantage of the a posteriori combination techniques, such as data fusion (Ceccherini et al. 2015) or the Kalman filter (Warner et al. 2014, Schneider et al. 2022) is to overcome these difficulties combining the Level 2 products supplied by the individual retrieval processors of the independent measurements.

The Complete Data Fusion (CDF) is an a posteriori method to combine atmospheric products from independent remote sensing observations of vertical profiles corresponding to nearby geolocations into a single estimate for a concise and complete characterization of the atmospheric state (Ceccherini et al. 2015). The CDF is termed 'complete' because it accounts for all features of the combined measurements, namely the retrieval errors of the fusing profiles and their correlations (represented by the covariance matrices, CMs) and the sensitivity of the retrieved profiles to the true profile (described by the averaging kernel matrices, AKMs), and can be regarded as a generalization of the weighted mean for cases where AKMs differ from the identity matrix. It has been proven (Ceccherini et al. 2015) that CDF provides results equivalent to those of the synergistic retrieval, when the linear approximation of the forward models is appropriate in the variability range of the individual retrievals results. The CDF has been applied since its introduction, about 10 years ago, to vertical profiles from both simulated and real measurements (one-dimensional analysis, 1D-CDF). During these years, the method has been developed and improved to extend its application to an increasingly large number of atmospheric products. In 2018, interpolation and coincidence errors were introduced in the analysis (Ceccherini et al. 2018) to overcome the quality degradation of the fused product encountered when the fusing profiles are either retrieved on different vertical grids or referred to different true profiles, i.e. not covering the same airmass. The upgraded algorithm was successfully used in the AURORA project (Cortesi et al. 2018) to fuse simulated measurements of Sentinels 4 and 5 (Tirelli et al. 2020, Zoppetti et al. 2021). As the CDF was limited to products of a single atmospheric variable, the generalization of CDF to Multi Target Retrieval (MTR, i.e. an inversion technique that enables the simultaneous retrieval of multiple atmospheric parameters from remote sensing measurements) products was performed in Tirelli et al. (2021). This formulation was used to combine products from FORUM and IASI-NG simulated measurements in case of perfect matching and of realistic mismatch, demonstrating the equivalence between the results obtained with the CDF and with the synergistic retrieval in both cases (Ridolfi et al. 2022).

In this study, the CDF algorithm, previously applied exclusively for 1D analysis, has been extended to two-dimensional products (2D-CDF) to evaluate the added value of combining coincident nadir and limb measurements. In the 1D analysis, the CDF's inputs are vertical profiles, unidimensional quantities mathematically represented by vectors. In the 2D analysis, the

CDF's inputs are two-dimensional fields consisting of vertical planes identified by the vertical dimension and the line of sight of the instrument. These 2D products can be of two types. The first type, is obtained by merging vertical profiles from 1D retrievals of independent along-track nadir measurements. Such fields provide vertically resolved information along a narrow swath and are characterized by correlations confined to the vertical dimension. The second type derives from tomographic retrievals which exploit multiple viewing geometries to reconstruct the atmospheric state. In this case, the retrieved fields are inherently smoothed and correlated in both the vertical and horizontal directions. In this work, we show an example of the synergy, for the ozone retrieval, between the Infrared Atmospheric Sounding Interferometer New Generation (IASI-NG), a nadir-looking sensor onboard the MetOp-SG-A satellite, and the Changing-Atmosphere Infrared Tomography (CAIRT), a candidate mission for ESA's Earth Explorer 11 program. The latter, proposed to fly in loose formation with MetOp-SG-A, represents a breakthrough in infrared limb sounding with its tomographic capabilities. CAIRT introduces a novel approach to limb sounding, leveraging imaging array detectors to perform atmospheric tomography from the middle troposphere to the lower thermosphere. Unlike limb scanners that retrieve single vertical profiles from individual scans, CAIRT's closely spaced acquisitions enable the reconstruction of 2D atmospheric cross-sections, capturing both horizontal and vertical variations with unprecedented spatial resolution. This capability, combined with IASI-NG's measurements, allows for advanced data fusion techniques like 2D-CDF to maximize the information extracted from these complementary datasets.

The paper is structured as follows: Section 2 outlines the mathematical foundations of the CDF method and its application to tomographic retrieval products. Section 3 describes the instruments and simulated measurements. Section 4 details the input calculations for the 2D-CDF, followed by performance assessment quantifiers in Sect. 5. Section 6 presents and discusses the results of the first 2D-CDF application to simulated measurements of ozone, including a comparison with the 1D limb/nadir combination. Finally, Sect. 7 draws the conclusions.

## 2 Method

# 2.1 Complete Data Fusion (CDF)

As discussed in the introduction, so far the CDF method has been developed to be applied to the fusion of vertical profiles, which in the mathematical formalism are represented with vectors. Therefore, the products in input to the algorithm are quantities unidimensional representing the value of an atmospheric parameter as a function of altitude. In case of limb measurements, it is possible to perform a two-dimensional retrieval in which the retrieved quantities are not vertical profiles but two-dimensional fields, where the two dimensions are associated with altitude and a coordinate along the line of sight of the instrument. For this reason, it can be useful to have a tool able to perform the data fusion of a set of two-dimensional fields retrieved from measurements of different instruments.

In this Section, we recall the equations of the CDF method and describe how to use it to perform the data fusion of a set of tomographic retrieval products. We use the formalism developed in Rodgers 2000.

## 2.2 CDF equations

We suppose to have performed the retrieval of N vertical profiles from remote sensing measurements with the optimal estimation method, using as a priori information the a priori profiles  $\mathbf{x}_{ai}$  and the a priori CMs  $\mathbf{S}_{ai}$ , where i assumes the values 1, 2, ...N. Therefore, we have obtained the retrieved profiles  $\hat{\mathbf{x}}_i$  that are characterized by the AKMs  $\mathbf{A}_i$  and by the CMs  $\mathbf{S}_i$ , given by:

$$\mathbf{A}_{i} = \left[ \mathbf{F}_{i} + \mathbf{S}_{2i}^{-1} \right]^{-1} \mathbf{F}_{i} \tag{1}$$

95 
$$\mathbf{S}_i = \left[\mathbf{F}_i + \mathbf{S}_{ai}^{-1}\right]^{-1},$$
 (2)

where

$$\mathbf{F}_i = \mathbf{K}_i^t \mathbf{S}_{vi}^{-1} \mathbf{K}_i \tag{3}$$

are the Fisher information matrices (Rodgers, 2000; Ceccherini et al., 2012), with  $S_{yi}$  the noise CMs of the observations (radiances) and  $K_i$  the Jacobians of the forward models calculated at the convergence point of the iterative retrieval process.

The retrieval CMs of Eq. (2) are the sum of two contributions, the noise CMs  $S_{n,i}$  and the smoothing error CMs  $S_{s,i}$ , which are given by:

$$\mathbf{S}_{\mathrm{n},i} = \left[\mathbf{F}_i + \mathbf{S}_{\mathrm{a}i}^{-1}\right]^{-1} \mathbf{F}_i \left[\mathbf{F}_i + \mathbf{S}_{\mathrm{a}i}^{-1}\right]^{-1} \tag{4}$$

$$\mathbf{S}_{s,i} = \left[ \mathbf{F}_i + \mathbf{S}_{si}^{-1} \right]^{-1} \mathbf{S}_{si}^{-1} \left[ \mathbf{F}_i + \mathbf{S}_{si}^{-1} \right]^{-1}. \tag{5}$$

The CDF of the N profiles is obtained minimizing the following cost function (Ceccherini et al., 2015):

$$\xi_{\text{CDF}}^{2}(\mathbf{x}) = \sum_{i=1}^{N} (\boldsymbol{\alpha}_{i} - \mathbf{A}_{i}\mathbf{x})^{t} \mathbf{S}_{\text{n},i}^{-1} (\boldsymbol{\alpha}_{i} - \mathbf{A}_{i}\mathbf{x}) + (\mathbf{x}_{\text{a}} - \mathbf{x})^{t} \mathbf{S}_{\text{a}}^{-1} (\mathbf{x}_{\text{a}} - \mathbf{x}) , \qquad (6)$$

where  $\mathbf{x}_a$  and  $\mathbf{S}_a$  are the a priori profile and CM used to constrain the fused profile and

$$\alpha_i = \hat{\mathbf{x}}_i - (\mathbf{I} - \mathbf{A}_i)\mathbf{x}_{ai} \,, \tag{7}$$

with **I** the identity matrix. The quantities  $\alpha_i$  defined by Eq. (7), in the linear approximation of the forward model, result independent of the a priori profiles  $\mathbf{x}_{ai}$  (Ceccherini, 2024), but maintain the dependence on the a priori CMs  $\mathbf{S}_{ai}$ .

The value  $\mathbf{x}_f$  for which  $\xi_{CDF}^2(\mathbf{x})$  is minimum provides the CDF solution:

$$\mathbf{x}_{\mathrm{f}} = \left(\sum_{i=1}^{N} \mathbf{A}_{i}^{t} \mathbf{S}_{\mathrm{n},i}^{-1} \mathbf{A}_{i} + \mathbf{S}_{\mathrm{a}}^{-1}\right)^{-1} \left(\sum_{i=1}^{N} \mathbf{A}_{i}^{t} \mathbf{S}_{\mathrm{n},i}^{-1} \boldsymbol{\alpha}_{i} + \mathbf{S}_{\mathrm{a}}^{-1} \mathbf{x}_{\mathrm{a}}\right), \tag{8}$$

which is characterized by the AKM and CM given by:

$$\mathbf{A}_{f} = \left(\sum_{i=1}^{N} \mathbf{A}_{i}^{t} \mathbf{S}_{n,i}^{-1} \mathbf{A}_{i} + \mathbf{S}_{a}^{-1}\right)^{-1} \sum_{i=1}^{N} \mathbf{A}_{i}^{t} \mathbf{S}_{n,i}^{-1} \mathbf{A}_{i}$$

$$(9)$$

115

$$\mathbf{S}_{\mathrm{f}} = \left(\sum_{i=1}^{N} \mathbf{A}_{i}^{t} \mathbf{S}_{\mathrm{n},i}^{-1} \mathbf{A}_{i} + \mathbf{S}_{\mathrm{a}}^{-1}\right)^{-1}.$$
(10)

A more general formula of the CDF that is applicable also when the noise CMs  $S_{n,i}$  are singular is given by (Ceccherini et al., 2022):

$$\mathbf{x}_{\mathbf{f}} = \left(\sum_{i=1}^{N} \mathbf{S}_{i}^{-1} \mathbf{A}_{i} + \mathbf{S}_{\mathbf{a}}^{-1}\right)^{-1} \left(\sum_{i=1}^{N} \mathbf{S}_{i}^{-1} \boldsymbol{\alpha}_{i} + \mathbf{S}_{\mathbf{a}}^{-1} \mathbf{x}_{\mathbf{a}}\right), \tag{11}$$

which is characterized by the AKM and CM given by:

$$\mathbf{A}_{f} = \left(\sum_{i=1}^{N} \mathbf{S}_{i}^{-1} \mathbf{A}_{i} + \mathbf{S}_{a}^{-1}\right)^{-1} \sum_{i=1}^{N} \mathbf{S}_{i}^{-1} \mathbf{A}_{i}$$

$$(12)$$

$$\mathbf{S}_{\mathrm{f}} = \left(\sum_{i=1}^{N} \mathbf{S}_{i}^{-1} \mathbf{A}_{i} + \mathbf{S}_{\mathrm{a}}^{-1}\right)^{-1}.$$
(13)

It is possible to demonstrate (Ceccherini, 2022; Ceccherini et al., 2022) that the CDF formula of Eq. (11) is equivalent to perform the data fusion with the Kalman filter (Kalman, 1960).

#### 2.3 Extension of the CDF to tomographic retrieval products

In the case of retrieval products that are two-dimensional fields, the retrieved atmospheric parameter (ozone for this study) is given on a two-dimensional grid, therefore, it can be represented by a matrix whose entries  $p_{jk}$  depend on two indices j and k which can vary from 1 to m and from 1 to n, respectively, every index being associated with one of the two dimensions. In the case of a two-dimensional product retrieved from a limb measurement, an index can be associated with altitude and the other index can be associated with the coordinate along the line of sight. Therefore, in this case the two-dimensional field can be seen as a set of vertical profiles, with each profile located at a different value of the coordinate along the line of sight. We can still use the formulas in Eq. (8) or Eq. (11), but we have to arrange the values on the grid of the field in the vector  $\hat{\mathbf{x}}_i$ . Figure 1 shows a schematic illustration of one-dimensional (1D) and two-dimensional (2D) state vectors used as input of the CDF. In the 2D case, the coloured subvectors of  $\hat{\mathbf{x}}$  represent vertical profiles associated with different along-track (ATK) positions, as schematically depicted on the right side of the figure. The simplest way to perform this arrangement is to fix one of the two indices, for example k, equal to the value 1 and fill the vector  $\hat{\mathbf{x}}_i$  by varying the other index j from 1 to m. Then, we fix the value of k equal to 2, and vary the index j from 1 to m. We repeat this procedure until we arrive to fix the value of k equal to

**Figure 1.** Schematic representation of state vectors  $\hat{\mathbf{x}}$  for 1D and 2D products input of the CDF algorithm.

n and vary the index j from 1 to m. In Eq. (14) we report this arrangement, where for simplicity we have omitted the index i, and, to save space, we have reported the transpose of the vector  $\hat{\mathbf{x}}$  (row vector) instead of the column vector.

$$\hat{\mathbf{x}}^{\mathbf{t}} = \begin{pmatrix} p_{11} & p_{21} & \dots & p_{m1} & p_{12} & p_{22} & \dots & p_{m2} & \dots & \dots & p_{1n} & p_{2n} & \dots & p_{mn} \end{pmatrix}$$
(14)

The arrangement of Eq. (14) implies an arrangement of the entries of the AKMs and CMs, indeed one entry of these matrices is characterized in the two-dimensional case by four indices. The entries of the AKM are given by:

$$A_{jkj'k'} = \frac{\partial \hat{p}_{jk}}{\partial p_{j'k'}},\tag{15}$$

where  $\hat{p}_{jk}$  is the retrieved parameter related to the two-dimensional grid point jk and  $p_{j'k'}$  is the true value of the parameter related to the two-dimensional grid point j'k'. The entries of the CM are given by:

$$S_{jkj'k'} = \langle (\hat{p}_{jk} - \langle \hat{p}_{j'k'} - \langle \hat{p}_{j'k'} - \langle \hat{p}_{j'k'} \rangle) \rangle \tag{16}$$

where  $\langle ... \rangle$  represent the mean values.

The arrangement of state vector described in Eq. (14) implies the following arrangement for the entries of the AKMs

and an analogous arrangement for the CMs.

The same arrangements have to be performed for the a priori two-dimensional field and the a priori CM. Once that the two-dimensional retrieved and a priori fields have been arranged in the way described by Eq. (14) and all the AKMs and CMs have been arranged in the way described by Eq. (17), the CDF can be performed using Eq. (8) or Eq. (11).

#### 155 3 Simulated measurements

## 3.1 Instruments

150

160

The Changing-Atmosphere Infra-Red Tomography Explorer (ESA 2025) is one of the two candidates for ESA's Earth Explorer 11. CAIRT aims to investigate the coupling between circulation (Butchart 2014) and composition in the middle atmosphere and to study their interaction with climate change (Domeisen and Butler 2020, Kidston et al. 2015). To do this, the 3-dimensional knowledge of the atmosphere, with high spatial resolution, is needed. From temperature, the information on the atmospheric gravity waves (Rhode et al. 2024), which drive the circulation, can be derived, while long-lived species provide information on the so-called age of air (Garny et al. 2024), which is the mean transport time from a reference surface, typically taken to be the tropical tropopause, to any given point in the stratosphere (Hall and Plumb 1994; Waugh and Hall 2002), and it serves as a proxy for changes in velocity of the circulation. The interactions of the middle atmosphere with the space environment above and with the troposphere below is investigated by measuring the downward flux of reactive nitrogen (NOy) from the

thermosphere (Funke et al. 2014), and by quantifying injection of pollutants and aerosol precursors into the upper troposphere and stratosphere (Pope et al. 2016; Khaykin et al. 2022).

CAIRT employs Fourier Transform Spectroscopy (FTS) to perform broadband measurements of thermal infrared radiation in the spectral range 718–2200 cm $^{-1}$  (13.93–4.55  $\mu$ m), with a spectral resolution of 0.39 cm $^{-1}$  for Norton Beer strong (NBS) apodised data. It operates in the limb-viewing geometry, providing high vertical resolution from approximately 4–5 km up to 115 km. A key innovation of CAIRT is its use of imaging array detectors that simultaneously capture limb emission spectra in two spatial dimensions—altitude and across-track—over a swath of about 400 km. This configuration allows for closely spaced (50 km) consecutive acquisitions along-track, achieving unprecedented horizontal resolution for limb sounders. CAIRT setup supports both day and night observations and enables the retrieval of multiple trace species across a wide vertical range.

IASI-NG (EUMETSAT 2025) is a nadir-viewing infrared sounder onboard the MetOp-SG A satellite, primarily designed to support Numerical Weather Prediction (NWP) at both regional and global scales. It provides high-resolution infrared radiance spectra, enabling the retrieval of temperature and humidity profiles with high accuracy (approximately 1 K and 5 % respectively), even in partly cloudy conditions. Secondary objectives include monitoring air quality by detecting tropospheric pollutants, and assessing climate–composition interactions through the observation of greenhouse gases. Like CAIRT, IASI-NG also uses FTS, covering a similar spectral range (645–2760 cm<sup>-1</sup>) with a spectral resolution of 0.25 cm<sup>-1</sup> (after NBS apodization) (Crevoisier et al. 2014, Andrey-Andrés et al. 2018). IASI-NG measurements are characterized by higher horizontal resolution but smaller vertical resolution with respect to CAIRT ones, and have information on the lower and middle troposphere. More precisely, IASI-NG covers a swath of about 2200 km including 14 fields of regard (FORs), each consisting of 16 instantaneous fields of view (FOVs) equivalent to circles of diameter of 12 km at nadir (see Fig. 2.3 in Smith and Crevoisier 2018). This allows IASI-NG to reach a spatial sampling of about 25 km both across-track (at least for the central FORs) and along-track.

In order to be able to combine CAIRT and IASI-NG measurements and to exploit the synergy, CAIRT, if selected, will fly in loose formation with MetOp-SG satellite, bringing on board IASI-NG together with several other instruments looking at nadir. The two satellites will fly on the same orbit, dephased of about 27° to match IASI-NG FOV with the region of the lines of sight of CAIRT closer to the surface. This will allow to exploit the complementary information of the two instruments, with CAIRT providing an unprecedented 3D view of the atmosphere from 4-5 km to 115 km, with high vertical and horizontal (both along and across track) resolution, and IASI-NG providing even better spatial resolution along and across track, allowing the extension of CAIRT measurements down to the surface.

## 3.2 Simulations

175

180

To evaluate the performance of the extended CDF algorithm, we conducted a series of simulations based on synthetic measurements from the CAIRT and IASI-NG instruments. These simulations were necessary due to the unavailability of IASI-NG and CAIRT real data. The simulated datasets reproduce realistic observational scenarios and allow us to test the algorithm under controlled conditions. In the following, we describe the simulation setup, including the forward models, instrument characteristics, and the atmospheric scenario used for generating the synthetic measurements. The CDF algorithm combines Level 2

products provided by the individual retrieval processors of the independent measurements. It is common for different Forward Models (FMs) to be used in the separate retrievals.

205

IASI-NG measurements have been simulated with the KLIMA (Kyoto protocol Informed Management of the Adaptation) code (Dinelli et al. 2023). The KLIMA code is a self-standing algorithm that can be used to simulate and to analyse the spectral radiance acquired by remote sensing measurements from observations in different geometrical configurations (limb, zenith and nadir) and spectral bands (from millimeter and sub-millimeter wavelengths to the near infrared). KLIMA is a lineby-line model and for these simulations we used the AER v3.8.1 spectroscopic database (Atmospheric and Environmental Research, Inc. Accessed 2025). In this study, it was used as the forward model to simulate ozone Jacobians for one of the Extended Reference Scenarios (ERS, Errera 2023) developed for the performance assessment of the CAIRT mission, within the CAIRT-SciRec (Science and Requirements Consolidation Study) project. In this paper, we focus on a single atmospheric scenario representative of daytime mid-latitude spring conditions (latitude 45°N, April), with the related ozone profile shown in Fig. 2. The ozone climatology used in this scenario was derived from simulations performed with the Whole Atmosphere Community Climate Model (WACCM). To simulate IASI-NG measurements, the high-resolution spectrum (0.005 cm<sup>-1</sup>), computed using the KLIMA code, was convolved with the Instrument Spectral Response Function (ISRF) of the IASI-NG instrument. Subsequently, the Noise Equivalent Spectral Radiance (NESR) and the Absolute Radiometric Accuracy (ARA) were added. The assumed characteristics of the IASI-NG instrument are those described previously (Sect. 3.1) and are based on Crevoisier et al. 2014 and Ridolfi et al. 2020. The simulated IASI-NG measurements assume an apodized spectrum using a Gaussian function with a Full Width at Half Maximum (FWHM) of 0.25 cm<sup>-1</sup>, matching the spectral resolution, and a spectral sampling step of 0.125 cm<sup>-1</sup>. The NESR is assumed to be half of the typical NESR of the current IASI instrument onboard MetOp (Crevoisier et al. 2014), while the ARA is specified to be better than 0.25 K ( $2\sigma$ ) when observing a blackbody at T = 280 K (see Figure 1 of Ridolfi et al. 2022).

On basis of the selected ERS atmospheric scenario, CAIRT limb-observations have been simulated by use of the Karlsruhe Optimized and Precise Radiative transfer Algorithm (KOPRA) (Stiller, 2000). The line-by-line radiative transfer model has been validated extensively over a long period (Glatthor et al., 1999; von Clarmann, 2002; von Clarmann et al., 2003; Tjemkes et al., 2003; Schreier et al., 2018; Höpfner and Emde, 2005; Griessbach et al., 2013). It is applied as the baseline forward model at Karlsruhe Institute of Technology and Instituto de Astrofísica de Andalucía (IAA-CSIC) for retrieval of atmospheric parameters from infrared limb-emission of the heritage instruments Michelson Interferometer for Passive Atmospheric Sounding (MIPAS)/Envisat, MIPAS-Balloon, MIPAS-aircraft, and the CAIRT demonstrators Gimballed Limb Observer for Radiance Imaging of the Atmosphere (GLORIA)-aircraft and GLORIA-balloon. CAIRT instrument specifications have been defined based on performance optimization during the ESA study CAIRT-SciReC, in the frame of which also these simulations have been performed. As an outcome of this, the CAIRT instrument specifications have been simulated by application of a vertical SEDF (System Energy Distribution Function) as a Gaussion function of 1.4 km FWHM to the KOPRA pencil-beam simulations. Further, in spectral dimension, the Norton-Beer-Strong apodization according to a maximum optical path difference of the CAIRT interferometer of 2.5 cm has been utilized. This results in a apodized spectral FWHM of 0.39 cm<sup>-1</sup>. The key radiometric requirements for CAIRT instruments are described in Sect. 4.4.4 of ESA 2025.

# 235 4 2D-CDF inputs calculation

In this study, the CDF algorithm was applied using the AKMs and CMs of IASI-NG and CAIRT, calculated as described in Sect. 2.2 from Eqs. (1,2) and (3). The Jacobian of IASI-NG,  $\mathbf{K}_{1\mathrm{D,I}}$ , was simulated as described in Sect. 3.2, and the noise CM of the observations (radiances),  $\mathbf{S}_{y\mathrm{I}}$ , was calculated for IASI-NG following the technical specifications of the instrument (see Sect. 3.1). For the tomographic analysis, we built a 2D Jacobian  $\mathbf{K}_{2\mathrm{D,I}}$  replicating  $\mathbf{K}_{1\mathrm{D,I}}$  for each along track position of the CAIRT bidimensional grid selected for the analysis. The 2D Jacobian for CAIRT  $\mathbf{K}_{2\mathrm{D,C}}$  was simulated as described in Sect. 3. The noise CM  $\mathbf{S}_{y\mathrm{C}}$  for CAIRT was calculated considering the ESA Noise-Equivalent Spectral Radiance (NESR) requirements described in the Mission Assumption and Technical Requirements (MATER) and Report for Assessment (Hoffmann, 2023). Finally, for CAIRT we calculated the 1D Jacobian  $\mathbf{K}_{1\mathrm{D,C}}$  from the 2D one, summing the elements of  $\mathbf{K}_{2\mathrm{D,C}}$  corresponding to the same altitude and to different ATK positions.

In the test cases of this study, we considered: a vertical grid with 61 altitude steps at 1 km intervals for the 1D data fusion, and a two-dimensional grid with a width of 800 km, height of 61 km, horizontal step of 50 km, and vertical step of 1 km for the tomographic data fusion (see Table 1). These choices were made according to the expected horizontal and vertical resolution of CAIRT.

| FUSION GRID | VERTICAL           | HORIZONTAL              |
|-------------|--------------------|-------------------------|
| 1D          | 0-61 km, 1 km step | /                       |
| 2D          | 0-61 km, 1 km step | 800 km wide, 50 km step |

Table 1. Vertical and horizontal specifications for 1D and 2D fusion grids.

The ozone climatology of McPeters and Labow (2012) was selected as a priori for CAIRT, IASI-NG and for the fused product. The a priori profile is shown in Fig. 2. The a priori error CM used for the 1D analysis is described by the following equation:

$$\mathbf{S}_{\mathbf{a},ij} = \sqrt{\mathbf{S}_{\mathbf{a},ij}\mathbf{S}_{\mathbf{a},ij}}e^{-\frac{|z_i - z_j|}{\text{lcorrz}}} \tag{18}$$

where  $z_i$  and  $z_j$  refer to the altitude values (i and j vary from 0 to 61), lcorrz is the correlation length in the vertical direction used to reduce oscillations in the retrieved profile. The a priori CM used for the 2D analysis is defined as:

$$\mathbf{255} \quad \mathbf{S}_{\mathbf{a},jkj'k'} = \sqrt{\mathbf{S}_{\mathbf{a},jkjk}} \mathbf{S}_{\mathbf{a},j'k'j'k'} e^{-\frac{\left|z_{j}-z_{j'}\right|}{\operatorname{lcorrh}}} e^{-\frac{\left|l_{k}-l_{k'}\right|}{\operatorname{lcorrh}}}$$

$$(19)$$

where lcorrh is the correlation length in the horizontal direction. Indices are the same as those described in Section 2.3, with j referred to the altitude, varying from 0 to 61, and k referred to the ATK position, varying from 0 to 21. In this study, we considered a vertical correlation length of 6 km, as this value is typically used for nadir ozone profile retrieval (Liu et al., 2010; Kroon et al., 2011; Miles et al., 2015), and a horizontal one of 25 km.

**Figure 2.** Ozone profile for the selected ERS scenario (TRUE; daytime, latitude 45°N, April) and a priori profile from McPeters and Labow climatology.

# 260 5 Quantifiers for CDF performance assessment

265

At the end of the CDF process, we carried out a quality evaluation analysis of the results. The quality assessment of the 1D and 2D fused products considered the following elements:

- the values of total error of CAIRT, IASI-NG and fused products, calculated as the square root of the diagonal elements
  of the related total CMs.
- the values of the diagonal elements of AKMs for CAIRT, IASI-NG and fused products.

In order to estimate quantitatively the quality improvement achieved by the CDF application with respect to the use of the individual products, we also calculated:

- the number of Degrees of Freedom (DOFs), a scalar measure of the number of independent quantities that can be measured, given by the trace of the AKM (Rodgers, 2000).
- the Shannon Information Content (SIC), defined as (Rodgers, 2000):

$$\Delta I_i = 0.5 * (\log_2 |\mathbf{S}_a| - \log_2 |\mathbf{S}_i|) \tag{20}$$

where  $|S_a|$  and  $|S_i|$  are the determinants of the CMs of the a priori and retrieved products. The SIC value provides the information gain obtained with the retrieval process with respect to the a priori information.

The performances of the synergy can also be evaluated in terms of the Synergy Factor (SF), a quantifier used to evaluate the synergy between two or more independent measurements. The SF is equal to 1 when the combined measurements are

complementary, and greater than 1 when a synergy between the two individual data sets really exists (supposing that the same a priori CM is used for the individual and fused measurements). To compare quantitatively the 1D-CDF and the 2D-CDF performances, we calculated the error  $SF(SF_{err})$  and the DOF  $SF(SF_{DOF})$ , as described below:

- the error synergy factor (SF<sub>err</sub>) (Aires, 2011). For each altitude level (j), the SF is defined as the ratio between the minimum total error of the fusing profiles  $(\sigma_{\text{tot},i}^{(j)})$  and the total error of the fused profile  $(\sigma_{\text{tot},f}^{(j)})$ 

$$\mathsf{SF}_{\mathrm{err}}^{(j)} = \frac{\min_{i=1,2,\dots,N} \sigma_{\mathrm{tot},i}^{(j)}}{\sigma_{\mathrm{tot},f}^{(j)}};\tag{21}$$

- the DOF synergy factor (SF<sub>DOF</sub>). For each altitude level (j), the SF is defined as the ratio between the diagonal element of the AKM of the fused profile  $(\operatorname{diag} A_f^{(j)})$  and the maximum diagonal element of the AKM of the fusing profiles  $(\operatorname{diag} A_f^{(j)})$ 

285 
$$\mathsf{SF}_{\mathrm{DOF}}^{(j)} = \frac{\mathrm{diag} \mathsf{A}_{f}^{(j)}}{\mathrm{max}_{i=1,2,\dots,\mathrm{N}} \mathrm{diag} \mathsf{A}_{i}^{(j)}}. \tag{22}$$

The synergy factors provide a quantitative evaluation of the improvement obtained in the fused product with respect to the most informative input product.

# 6 Results

295

300

The aim of this study is to demonstrate the feasibility of the CDF application to tomographic retrieval products and to compare
the results obtained in selected 1D and 2D test cases. In particular, we applied the 1D-CDF and 2D-CDF algorithm to simulated
ozone measurements of two sensors with the specifications of IASI-NG and of the CAIRT mission instrument operating at nadir
and limb, respectively. We consider three test cases:

- CASE 1: 1D-CDF is used to combine one ozone profile derived from the analysis of a single CAIRT acquisition and four IASI-NG ozone simulated measurements in coincidence. This case represents the application of the CDF to two 1D products from sensors with the specifications mentioned above.
- CASE 2: 2D-CDF is used to combine a 2D ozone field derived from the analysis of a single CAIRT acquisition and a 2D ozone field obtained from 84 IASI-NG simulated measurements on the same along-track grid, with step 50 km (four IASI-NG measurements for each ATK positions). The difference with case 1 is that the 2D configuration allows to consider the horizontal variability of the atmosphere through the combination of a single limb measurement and a set of nadir measurements overlapping the lines of sight of the limb one.

– CASE 3: 2D-CDF is used to combine the 2D ozone fields derived from the analysis of 51 CAIRT acquisitions (spaced 50 km apart) and a 2D ozone field derived from 84 IASI-NG simulated measurements (as in case 2) on the same along-track grid with step 50 km. In this test case, we finally represent the configuration expected to be implemented for the limb measurements of the CAIRT mission and for the nadir measurements of IASI-NG.

The geometric representations of Case 1, Case 2, and Case 3 are shown in Fig.3, Fig. 5 and 8, respectively. It is important to note that:

- at each ATK position of the 2D grid, four IASI-NG simulated measurements (the ones in the 50 km x 50 km region close to the along-track position) were selected for the data fusion process to properly take into account the spatial resolution of IASI-NG for small off-nadir angles (see Sect. 3.1);
- cases 1 and 2 are designed as a basis of comparison with the results obtained in case 3. To this aim, data of cases 1 and 2 are simulated considering the same instrumental specifications for the two sensors involved in case 3, but since only one CAIRT acquisition is used for the retrieval, the information contained in CAIRT simulated measurements is not fully exploited. Case 1 further assumes the retrieval of only 1 profile under the hypothesis of homogeneous atmosphere.

As described in Sect. 4, for the 1D data fusion, we considered a vertical grid of 61 altitude levels with 1 km steps according to the expected vertical resolution of CAIRT. For the 2D data fusion, we considered a two-dimensional grid extended 800 km wide and 61 km high with horizontal steps of 50 km and vertical steps of 1 km according to CAIRT expected horizontal and vertical resolutions (see Table 1). We analyzed, for each case, the improvements in the ozone retrieval (in terms of diagonal elements of the AKMs, total errors, SIC values, and number of DOFs) coming from the exploitation of the synergy between CAIRT observations and several nadir measurements along the CAIRT line of sight, comparing the data fusion results to those obtained for the individual products of CAIRT and IASI-NG measurements.

**Figure 3.** Geometric schematic of Case 1, illustrating a single CAIRT limb-viewing acquisition together with four overlapping IASI-NG nadir measurements (1D case).

# 6.1 Case 1

330

335

**Figure 4.** Case 1. Diagonal elements of the AKMs (left), a priori (from McPeters and Labow climatology, see Sect.4) and total errors (right) for IASI-NG, CAIRT and the fused product.

In the first test case (see Fig. 3), we reproduced the 1D retrieval from a limb acquisition, assuming that the atmosphere sounded by CAIRT is horizontally homogeneous. We applied the CDF in its 1D formulation to one CAIRT and four IASI-NG vertical profiles of ozone analyzing the results for a limb-nadir 1D combination. Figure 4 illustrates the results of case 1 for the vertical profiles of the AKM diagonal elements and total errors. The profile of the AKM diagonal elements of the fused product is superimposed on that of CAIRT from approximately 8 to 60 km, while in the lower troposphere it shows higher values with respect to both CAIRT and IASI-NG. The shape of the profile of the AKM diagonal elements for the fused product is largely governed by CAIRT measurements in the 8–60 km altitude range. This dominance is due to the limb observation geometry, which provides higher vertical resolution and benefits from longer atmospheric path lengths at the expense of the horizontal resolution. These two factors explain the higher information content of the CAIRT product compared to that of IASI-NG.

Near the surface, the fused product has the same values of the IASI-NG product, as expected by the fact that CAIRT does not perform measurements below 5 km. The highest level of vertical sensitivity is shown in the region between 8 and 20 km, where the values of the AKM elements vary from 0.4 to 0.75, reaching a peak at 15 km. The total error profile of the fused product is overlapped to that of CAIRT and more than 50 % smaller than that of IASI-NG from roughly 8 km up to 60 km. Below 8 km the fused product displays the highest quality, with respect to the individual products. The a priori error profile is also reported in the plot: it shows some oscillations (larger errors are found in correspondence of altitudes where the profile has a large slope) that are transferred in the total error profile of both the single products and the synergistic one.

Table 2 shows the values obtained for the number of DOFs and SIC for CAIRT, IASI-NG and the fused product. The fused product shows higher values of the two quality quantifiers with respect to the individual products, in particular with respect to IASI-NG (a gain of 3 times for the SIC and 4.5 times for the number of DOF), as expected.

| Product | DOF   | SIC   |
|---------|-------|-------|
| CAIRT   | 26.60 | 70.62 |
| IASI-NG | 5.97  | 23.07 |
| FUSED   | 27.25 | 75.05 |

Table 2. Case 1. DOFs and SIC values for CAIRT, IASI-NG and the fused product.

#### 6.2 Case 2

**Figure 5.** Geometric schematic of Case 2, illustrating one CAIRT limb-viewing acquisition together with multiple IASI-NG nadir measurements (2D case).

In the second case (see Fig. 5), we considered the two-dimensional grid previously described (see Table 1). We applied the CDF to a 2D ozone field obtained from one CAIRT acquisition and a 2D ozone field obtained from 84 measurements of IASI-NG, i.e. 4 nadir profiles for each of the 21 ATK positions in the horizontal dimension of the grid. In this case, the results for the diagonal elements of the AKM and for the total errors are provided for the two-dimensional grid in the maps of Fig. 6 and for the central ATK position in Fig. 7, showing the profiles as in case 1. In this case study, the information coming from the measurements of the two sensors is not homogeneous in the horizontal plane, but condensed around the central ATK positions (the CAIRT instrument is supposed to be on the right-hand side). From the maps of Fig. 6 it is easy to infer that the fused product ensures the highest level of information with the lowest level of total error. A single measurement of CAIRT shows a limited potential to cover the grid points, as the non-zero elements of the AKM are centered in a small part of the horizontal plane, where the tangent points of the line of sight are grouped. The characteristic shape of the information content in CAIRT measurements, shown in the middle-left panel, is primarily determined by the limb-viewing geometry. The information is concentrated along the lines of sight, reaching its maximum at the tangent points. The resulting arc-shaped pattern reflects the increase in line-of-sight altitude as the distance from the tangent point grows. Moreover, the tangent altitude shifts closer to the satellite with increasing altitude, which explains the higher averaging kernel values observed at positive ATK distances.

| Product | DOF    | SIC    |
|---------|--------|--------|
| CAIRT   | 29.02  | 66.14  |
| IASI-NG | 117.82 | 452.41 |
| FUSED   | 137.99 | 495.34 |

Table 3. Case 2. DOFs and SIC values for CAIRT, IASI-NG and the fused product.

The values of the number of DOFs and SIC are summarized in Table 3. For the 2D cases, it is important to underline that these values refer to a grid of 1281 points, i.e. 21 horizontal profiles with 61 vertical levels. The fused product shows the highest number of DOFs and SIC value and IASI-NG, in this particular case, demonstrates the greatest contribution to the fused product information content as its measurements cover the whole 2D grid space. The DOFs and SIC values for CAIRT are similar to those of case 1.

The results obtained for the profiles corresponding to the central ATK position are shown in Fig. 7. The profile of the AKM diagonal elements of the fused product shows values greater than 0.25 between 8 and 20 km (with a peak of 0.35 at 10 km), demonstrating greater vertical sensitivity with respect to the individual products. CAIRT and IASI-NG have the same behavior and nearly the same values between 15 and 50 km, although IASI-NG shows higher values from 0 to 15 km (with a maximum value of 0.27 near 10 km) and CAIRT above 50 km, closer to those of the fused product profile. The total error of the synergistic product is the smallest, with a maximum value of 0.3 ppmv at 30 km. In the vertical range between 8 and 50 km, the total errors of the fused product profile are up to 0.1 and 0.2 ppmv smaller than those of IASI-NG and CAIRT, respectively. The individual products are very close to the synergistic one in the remaining part of the vertical range (CAIRT in the upper part and IASI-NG near the surface).

# 370 **6.3** Case 3

In case three (see Fig. 8), multiple CAIRT measurements were combined on the 2D grid to simulate the retrieval configuration that could be obtained from consecutive acquisitions across the same region, as well as a number of IASI-NG measurements for each point of the ATK grid according to the corresponding spatial resolution (as in case 2). We selected an atmospheric region on the plane of the line of sight of the CAIRT acquisitions, and considered all the consecutive CAIRT measurements (resulted to be 51) that have a Jacobian significantly different from zero in this region. From these 51 Jacobians and the noise CMs of the radiances obtained by the specifications of the two instruments described in Sect. 3, we calculated the AKMs and CMs as described in Sects. 2.2 and 2.3, using the a priori information described in Sect. 4. In Figs. 9 and 10 we show the results of these calculations. In Figure 9, it was not possible to maintain the same colour scale values for the total error, as some features would have been lost. It is worth noting that the total error of IASI-NG is equal to that of case 2 in Fig. 6. Through the analysis of Fig. 9 and the comparison with Fig. 6, it is evident how the exploitation of a tomographic configuration improves CAIRT performances and consequently those of the synergistic product in terms of both spatial resolution (larger values of the AK) and smaller total error. The synergistic product of limb and nadir measurements provides the highest level of information

**Figure 6.** Case 2. Diagonal elements of the AKMs (left column) and total errors (right column) in the CAIRT 2D retrieval grid for (from the top) IASI-NG, CAIRT and the fused product.

Figure 7. Case 2. Diagonal elements of the AKMs (left), a priori and total errors (right) for IASI-NG, CAIRT and fused products for the central ATK position.

**Figure 8.** Geometric schematic of Case 3, illustrating multiple CAIRT limb-viewing acquisitions together with multiple IASI-NG nadir measurements (2D tomographic case).

and the lowest total error over the two dimensional grid. The vertical region which is interested by the highest improvement in terms of information gain (due to the exploitation of the limb/nadir synergy) is the Upper Troposphere Lower Troposphere (UTLS) region, from 8 to 20 km. In contrast to case two, the two fields — AKM diagonal elements and total errors — are horizontally homogeneous, as a result of the strong overlap in lines of sight between consecutive acquisitions. Thus, the results shown in Fig. 10 are exactly the same for all the 21 ATK positions. The profile of the AKM diagonal elements for the fused product reaches a peak of 0.6 around 15 km, with values greater than 0.4 between 8 and 20 km. The CAIRT profile shows the same behavior as the fused one, with lower values, and no information content for the altitude levels below 4 km since CAIRT does not perform measurements in this region. In these layers, IASI-NG measurements demonstrate, as expected, a sounding capability complementary to that of CAIRT. The total error peak values (around 30 km) vary in a range between 0.22 (fused

| Product | DOF    | SIC    |
|---------|--------|--------|
| CAIRT   | 307.44 | 642.46 |
| IASI-NG | 117.82 | 452.41 |
| FUSED   | 366.49 | 949.66 |

**Table 4.** Case 3. DOFs and SIC values for CAIRT, IASI-NG and the fused product.

product) and 0.4 (IASI-NG) ppmv, with CAIRT maximum value of about 0.33 ppmv. The total error of CAIRT differs by 0.1 ppmv from the synergistic one from 8 km up to 50 km. For IASI-NG, the difference with the error profile of the fused product is similar to that of CAIRT from 8 to 25 km, then it begins to increase up to 60 km.

The values of the number of DOFs and SIC for case 3 are summarized in Table 4. The fused product shows the highest number of DOFs and SIC values compared to the individual products. In this case, the 51 measurements of CAIRT cover the whole 2D space and demonstrate the highest contribution to the fused product. For IASI-NG, DOFs and SIC values are exactly the same of case 2, as expected.

## 6.4 1D versus 2D comparison

As described in Sect. 5, the performances of the synergy can be evaluated in terms of the the synergy factors SF<sub>DOF</sub> and SF<sub>err</sub>. Figure 11 shows the results for the two SFs, comparing the SF profiles for the 1D and 2D (case 3) data fusion studies. For 1D analysis, it is possible to take advantage of the synergy only below 8 km (SF<sub>err</sub> and SF<sub>DOF</sub> greater than 1), in the remaining altitude range only the complementarity between the two measurements can be exploited (SF<sub>err</sub> and SF<sub>DOF</sub> equal to 1). For the tomographic analysis, the synergy is fully exploited over the whole altitude range. The error SF once again demonstrates that the highest improvement from the synergy is obtained in the altitude range from 10 to 20 km.

# 6.5 2D resolution

The spatial resolution is an important quantity to characterize a retrieval. In this study we derived the spatial resolution for the tomographic retrieval from the 2D AKM (see Eq. (17). The resolution can be defined from the rows of the AKM which describe how the estimate of the retrieved state vector, at a fixed ATK distance and altitude, is affected by a perturbation of the true state in each point of the retrieval grid. In particular, the resolution for each point of the grid (i.e. 1281 points) is calculated as the Full Width Half Maximum (FWHM) of the corresponding averaging kernels (Rodgers 2000, Worden et al. 2004). Figure 12 shows four maps for the 2D averaging kernels (case 3) corresponding to the ozone retrieved value obtained from the data fusion for the central ATK position (0 km) and different altitudes: 8, 15, 25 and 35 km (from top left clockwise). The black contour on the images shows where the averaging kernel is at 50% of the maximum value of the selected AKM row, i.e the value from which we can calculate the FWHM. It is easy to evaluate the resolution in the vertical and horizontal directions from Fig. 12: at 8 km the vertical resolution is approximately 2.7 km and the horizontal close to 60 km, not symmetric in the vertical direction; at 15 km the spatial resolution reaches the maximum with less than 2 km vertical and nearly 60 km in the

**Figure 9.** Case 3. Diagonal elements of the AKMs (left column) and total errors (right column) in the CAIRT 2D retrieval grid for (from the top) IASI-NG, CAIRT and the fused product.

**Figure 10.** Case 3. Diagonal elements of the AKMs (left), a priori and total errors (right) for IASI-NG, CAIRT and fused products for a generic ATK position.

Figure 11. Comparison between 1D and 2D cases of DOF SF (left) and error SF (right).

425

horizontal direction; at 30 km and 35 km the vertical resolution is 2.25 km and 2.5 km respectively, while the horizontal one is roughly 100 km for both, but with a higher value at 35 km.

In Fig. 13, we show the vertical and horizontal resolutions for the whole vertical range, for the fused product, IASI-NG and CAIRT (case 3). The vertical resolution is calculated from the matrix:

$$A_{jk_0,j'k_0} = \frac{\partial \hat{p}_{jk_0}}{\partial p_{j'k_0}} = B_{jj'}(k_0), \tag{23}$$

a sub-matrix derived from the 2D AKM (see Eq. (17)) that refers to the sensitivity of the retrieved state vector, for a selected altitude j and fixed ATK position  $k_0$ , to the true state at different altitudes j' and same ATK position  $k_0$ . The rows of  $B_{jj'}(k_0)$  are the vertical averaging kernels; consequently, the vertical resolution is calculated as the FWHM of these averaging kernels.

The horizontal resolution is calculated from the matrix:

435

$$A_{jk_0,jk'} = \frac{\partial \hat{p}_{jk_0}}{\partial p_{jk'}} = C_{k_0k'}(j), \tag{24}$$

a sub-matrix derived from the 2D AKM (see Eq. (17)) that refers to the sensitivity of the retrieved state vector, for a selected altitude j and fixed ATK position  $k_0$ , to the true state at the same altitude j and for different ATK positions k'. The rows of  $C_{k_0k'}(j)$  are the horizontal averaging kernels, consequently the horizontal resolution is calculated as the FWHM of these averaging kernels.

**Figure 12.** Maps of the 2D AKM rows that correspond to the sensitivity of the retrieved vector to the true state at the central along track position and different altitudes: 8 km, 15 km, 25 km and 35 km from top left clockwise. Results are shown for the fused product.

In Figure 13, we show that the vertical resolution profile of the fused product nearly overlaps with that of CAIRT from 10 to 60 km with values of about 2-3 km, and has lower values near the surface. CAIRT maintains a better vertical resolution from 4 to 8 km. IASI-NG shows its best vertical resolution near the surface and up to 15 km with values in a range from 5 to 7 km. The horizontal resolution values of IASI-NG are, as expected for a nadir product, the best one: 50 km from 5 to 30 km degrading to 100 km at the top of atmosphere and near the surface. As we said above, the results reported in Fig. 13 refer to the resolution analysis for case 3 (the 2D case that we use as reference). For this reason, the horizontal resolution of IASI-NG

Figure 13. Vertical (left) and horizontal (right) resolutions for IASI-NG, CAIRT and the fused product.

shown in Fig. 13 is not that expected from its specifications (25 km), but that which derives from the data fusion process (50 km) on the 2D grid where, as we said above, four IASI-NG measurements are fused for each ATK position. The horizontal resolution of IASI-NG shown in Fig. 13 does not reflect the nominal resolution expected from the instrument specifications, which indicate a 25 km spacing between adjacent measurements. This is because, in the data fusion process, four IASI-NG measurements were combined for each ATK position, resulting in a minimum achievable horizontal resolution of 50 km.

The fused profile exploits the synergy between nadir and limb measurements showing horizontal resolution values from 100 to 70 km below 10 km, closer to 70 km from 10 to 20 km, lower than 100 km until 45 km and degrading to 150 km up to 60 km of altitude. CAIRT values of horizontal resolution are greater than 200 km under 8 km of altitude, vary from 100 to 75 km in the range of 10 to 20 km, and oscillate from 100 to 180 km in the remaining vertical range.

## 7 Conclusions

450

In this study, we extended the CDF algorithm, previously applied only to 1D products, to tomographic products (2D-CDF). This new approach allows for maximizing the information extracted from complementary nadir/limb or limb/limb datasets. We applied the 2D-CDF to simulated ozone products from the nadir sensor IASI-NG, flying aboard the MetOp-SG satellite, and the limb sensor CAIRT, a candidate for the ESA Earth Explorer 11 program. If selected, CAIRT will fly in loose formation with MetOp-SG, enhancing the synergy between the two sensors. CAIRT is expected to introduce a novel approach to limb sounding, leveraging its tomographic capabilities to explore the potential of advanced data fusion techniques, like 2D-CDF. We demonstrated the feasibility of applying the CDF algorithm to tomographic retrieval products and compared the results obtained in three selected test cases. Case 1 represents the application of the CDF to two 1D products from sensors with the instrumental specifications of IASI-NG and CAIRT, under the hypothesis of homogeneous atmosphere. Case 2 allows to consider the horizontal variability of the atmosphere through the combination of a single limb measurement and a set of nadir measurements overlapping the lines of sight of the limb one. This is the first application of the 2D-CDF, but since only one

CAIRT acquisition is used for the retrieval, we do not fully exploit the information contained in the CAIRT measurements. In case 3, we finally represent the tomographic configuration expected to be implemented for the acquisitions of the CAIRT mission, thus applying the CDF to a 2D tomographic case. For each case, we evaluated the performance of the data fusion process in terms of the number of degrees of freedom, the Shannon information content, the total errors, and the diagonal elements of the AK. From the analysis of these quality quantifiers, it is evident that the tomographic configuration represented in case 3 allows to improve significantly the performances of the synergistic product in terms of larger values of the AK, smaller total error and increased information gain, especially in the UTLS region, from 8 to 20 km. Furthermore, we calculated the error and DOF synergy factors to quantitatively compare the 1D-CDF and the 2D-CDF (case 3) performances. The results show that for the 1D analysis it is possible to take advantage of the synergy only below 8 km while for the tomographic case, the synergy is fully exploited over the whole altitude range. To complete the analysis of the 2D tomographic case, we derived the spatial resolution from the 2D AKMs obtained in case 3. The results show that the fused profile includes the information coming from the high horizontal resolution of the nadir measurements and that coming from the high vertical resolution of the limb measurements, fully exploiting the synergy between them.

# Data availability

Test data presented are available upon request to the authors.

## **Author contributions**

CT implemented the 2D-CDF algorithm, computed the CDF solution for the presented test cases and wrote the paper's draft version. SC developed the theoretical background for the CDF (1D and 2D) and contributed to the paper's draft version. CT and SC selected the test cases and interpreted the results. PR proposed and coordinated the study, contributed to the study planning and to the interpretation of the results. SDB performed the calculation of the IASI-NG forward model Jacobian. MH and BF performed the calculation of the CAIRT forward model Jacobian. UC coordinated the IFAC team working in this study.

All the authors reviewed the paper's manuscript.

## **Competing interests**

The authors declare that they have no conflict of interest.

# **Financial support**

The methodology described in this paper was developed in the frame of the CAIRT-SciReC study, contract ESA/ESTEC no. 4000136480/21/NL/LF, aimed to consolidate the scientific requirements of CAIRT mission, candidate for ESA Earth Explorer 11.

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
