# Peer review of "Extension of the Complete Data Fusion algorithm to tomographic retrieval products"

_EGUsphere, 2025_

## Author Comment (AC1)

**Reviewer 1**

\*\*\*\*\*\*\*\*\*\*\*\*\*\*\*\*\*\*\*\*\*\*\*\*\*\*\*\*\*\*\*\*\*\*\*\*\*\*\*\*\*\*\*\*\*\*\*\*\*\*\*\*\*\*\*\*\*\*\*\*\*\*\*\*\*\*\*\*\*\*\*\*\*\*\*\*\*\*\*\*\*\*\*\*\*

We thank the reviewer for the useful comments. In the following, we answer the specific comments (included in **"boldface"** for clarity) and, whenever required, we describe the related changes implemented in the revised manuscript. Page and line numbers indicated refer to the revised version of the paper. The text added in the revised version of the paper to address the reviewers' comments is highlighted in red.

\*\*\*\*\*\*\*\*\*\*\*\*\*\*\*\*\*\*\*\*\*\*\*\*\*\*\*\*\*\*\*\*\*\*\*\*\*\*\*\*\*\*\*\*\*\*\*\*\*\*\*\*\*\*\*\*\*\*\*\*\*\*\*\*\*\*\*\*\*\*\*\*\*\*\*\*\*\*\*\*\*\*\*\*\*

- **Abstract and introduction (especially lines 32-33): The statement that CDF has been "used so far to combine only one-dimensional atmospheric products (vertical profiles) from simultaneous and independent remote sensing observations of the same air mass" might be confusing, given that several 1D profiles within a certain spatiotemporal extent (e.g. L3 data grid) can be fused, so (exactly) the same air mass sounds too strong. Brief Section 2.1 is clearer on this and might therefore rather be part of the introduction.**

We thank the Reviewer for pointing out this possible source of confusion. In the introduction we changed the reported sentence using the expression "vertical profiles corresponding to nearby geolocations" . In the abstract we modified the text consistently  (lines 4-6).

We modified the Introduction to improve clarity and contextualization. These modifications are visible in the revised version.

- **With that, the difference should be explained between the tomographic retrieval and an along-track curtain of nadir observations, in terms of retrieval and its information characteristics (i.e. the former having vertical and horizontal information smoothing and covariances).**

We thank the Reviewer for this helpful suggestion. In the revised manuscript, we have added a short explanatory passage in the Introduction (lines 55–62) to clarify the differences between a tomographic retrieval and an along-track curtain of nadir observations.

Added text (55-62) : In the 1D analysis, the CDF's inputs are vertical profiles, unidimensional quantities mathematically represented by vectors. In the 2D analysis, the CDF's inputs are two-dimensional fields consisting of vertical planes identified by the vertical dimension and the line of sight of the instrument. These 2D products can be of two types. The first type, is obtained by merging vertical profiles from 1D retrievals of independent along-track nadir measurements. Such fields provide vertically resolved information along a narrow swath and are characterized by correlations confined to the vertical dimension. The second type derives from tomographic retrievals which exploit multiple viewing geometries to reconstruct the atmospheric state. In this case, the retrieved fields are inherently smoothed and correlated in both the vertical and horizontal directions.

- **Section 6: Having subsections for the discussion of the three cases would improve readability, but, more importantly, schematic drawings of the spatial configuration of the measurements that are combined in the three cases would provide a better understanding.**

We added the subsections (Case1, Case2, Case3) and the schematic drawings for each case (Figures 3, 5, 8).

- **Moreover, case 1 does not seem to be fully representative, not only because of its dimensionality, but also as the fused product is fully dominated by CAIRT (Figure 1). This should be appropriately discussed (instead of being just mentioned in lines 287-288), also in view of the DFS differences between Tables 1 and 2, which are minor for CAIRT (how?) but substantial for IASI-NG.**

As described in lines 291-293, Case 1 is designed to provide a basis of comparison with the 2D results. Case 1 results are described reporting the same quantifiers in order to be able to fully understand the improvement in the different aspects involved in the comparison analysis.

We added a paragraph in the text (line 327-330) to clarify why the fused product is fully dominated by CAIRT:

The shape of the profile of the AKM diagonal elements for the fused product is largely governed by CAIRT measurements in the 8–60 km altitude range. This dominance is due to the limb observation geometry, which provides higher vertical resolution and benefits from longer atmospheric path lengths at the expense of the horizontal resolution. These two factors explain the higher information content of the CAIRT product compared to that of IASI-NG.

In Case 1 the information from one CAIRT acquisition is gathered in a 1D profile, while in Case 2 the information from the limb acquisition covers the entire area of CAIRT lines of sight. Different a priori covariance matrices are used in the two cases to consider, in the 2D case, the horizontal information smoothing. This leads to small differences in CAIRT DOFs and SIC values for the two cases. In Case 1 only one IASI-NG measurement is considered while in Case 2 the IASI-NG measurements are 84. This fact explains the substantial difference between the DOFs and SIC values for the two cases, for IASI-NG.

- **It seems inappropriate and misleading to indicate the fused DFS to have a 'gain' or X times higher DFS than the individual products, as this largely depends on the number of profiles involved, rather than on the fusion performance.**

In Case 1, we fused one IASI-NG and one CAIRT observation, so we did not modify the sentence describing the DFS and SIC values reported in Table 1. For case 3, we modified the text accordingly to the reviewer's request (lines 395-398):

The fused product shows the highest number of DOFs and SIC values compared to the individual products. In this case, the 51 measurements of CAIRT cover the whole 2D space and demonstrate the highest contribution to the fused product. For IASI-NG, DOFs and SIC values are exactly the same of case 2, as expected.

- **The CAIRT AKM diagonal in Figure 2 (middle-left) looks interesting but lacks explanation. Could you elaborate on the 'smile' shape of the information between 10 and 20 km, and on the offset of the vertical information towards positive along-track distances (closer to CAIRT)?**

We added a description of the 'smile' shape and of the results showed in figure 6 in general (lines 351-355):

The characteristic shape of the information content in CAIRT measurements, shown in the middle-left panel, is primarily determined by the limb-viewing geometry. The information is concentrated along the lines of sight, reaching its maximum at the tangent points. The resulting arc-shaped pattern reflects the increase in line-of-sight altitude as the distance from the tangent point grows. Moreover, the tangent altitude shifts closer to the satellite with increasing altitude, which explains the higher averaging kernel values observed at positive ATK distances.

- **It could then be clarified from the configuration difference why these features are missing in Figure 4 for the third case. Here, the color scale should be updated to avoid saturation. Why keep the same color scale for the AKM diagonals between Figures 2 and 4, and not for the errors, wherefrom it would be clearer that the IASI-NG total errors in Figures 2 and 4 are the same?**

The combination of strongly overlapping lines of sight in consecutive acquisitions results in horizontally homogeneous information content in Fig. 9 (case 3). To address this comment, we have added a clarifying sentence at lines 385–386: In contrast to case two, the two fields — AKM diagonal elements and total errors — are horizontally homogeneous, as a result of the strong overlap in lines of sight between consecutive acquisitions.

The variability of the error is greater than that of the AKM. As a result, when using the same colour scale for the maps corresponding to both cases, certain features may not be clearly visible. We added a sentence in the text that explain our choice and underline the fact that the total error maps for IASI-NG are the same for the two cases (lines 378-379): In Figure 9, it was not possible to maintain the same colour scale values for the total error, as some features would have been lost. It is worth noting that the total error of IASI-NG is equal to that of case 2 in Fig. 6.

- **Lines 338-339: "Thus, the results shown in Fig. 5 are the same for all the 21 ALT positions." Is this exactly the same, or by approximation?**

The results shown in Fig. 5 are exactly the same for all the 21 ALT positions.

- **Please provide a physical or information-wise interpretation of the synergy factors defined in Equations (21) and (22).**

We added the sentence (lines 286-287):

The synergy factors provide a quantitative evaluation of the improvement obtained in the fused product with respect to the most informative input product.

- **Lines 396-400: Although an attempt is made to explain the origin of the IASI-NG horizontal resolution estimates, this is not clear from the current description.**

We rephrased the text describing the horizontal resolution of IASI-NG (lines 439-442):

The horizontal resolution of IASI-NG shown in Fig. 13 does not reflect the nominal resolution expected from the instrument specifications, which indicate a 25 km spacing between adjacent measurements. This is because, in the data fusion process, four IASI-NG measurements were combined for each ATK position, resulting in a minimum achievable horizontal resolution of 50 km.

**Technical corrections:**

- **Line 19: Referring only to Aires (2011, 2012) here seems too limited.**

The idea of the first part of the introduction is to first introduce the reference to two papers (i.e., Aires 2011, 2012) with a theoretical description of how to exploit the synergy between remote sensing measurements. In the next sentence we introduce the references (nearly 20) to papers about different types of applications. We added here some new references (Cortesi et al. 2016, Sofieva et al. 2022, Staelin et al. 1995).

If the reviewer has a suggestion for other theoretical papers, we will add them to Aires 2011, 2012.

- **Line 44: The meaning of "referred to different true profiles" is not clear here. Does this mean "not covering exactly the same air mass" here?**

Yes, this is the meaning. We changed the sentence in order to clarify: In 2018, interpolation and coincidence errors were introduced in the analysis (Ceccherini et al. 2018) to overcome the quality degradation of the fused product encountered when the fusing profiles are either retrieved on different vertical grids or referred to different true profiles, i.e. not covering the same airmass.

- **Appropriate referencing is missing in Section 3.1.**

We added the references as suggested by the reviewer.

- **Lines 148-149: To my understanding, the age of air does not equal "the time needed by a tropospheric air parcel to reach the stratosphere"**

We agree that the definition of the age of air used in the text can be improved.

Garny et al., 2024 (https://agupubs.onlinelibrary.wiley.com/doi/full/10.1029/2023RG000832) writes: A common way to diagnose the strength of the overall transport circulation in the stratosphere is by the mean transport time from a reference surface, typically taken to be the tropical tropopause, to any given point in the stratosphere (Hall & Plumb, 1994; Waugh & Hall, 2002). Those mean transport times are referred to as the mean age of stratospheric air.

We have modified the sentence strictly using Garny et al., 2024 definition (lines 160-164):

From temperature, the information on the atmospheric gravity waves (Rhode et al. 2024), which drive the circulation, can be derived, while long-lived species provide information on the so-called age of air (Garny et al. 2024), which is the mean transport time from a reference surface, typically taken to be the tropical tropopause, to any given point in the stratosphere (Hall and Plumb 1994; Waugh and Hall 2002), and it serves as a proxy for changes in velocity of the circulation.

- **Line 196: "monochromatic radiance" There should be several of these at least? Please explain.**

We made a mistake: "radiance" should be "radiances". We reformulated the sentence, adding the information about the resolution (lines 210-211):

In order to simulate IASI-NG measurements, the high-resolution spectrum (0.005 cm−1), computed using the KLIMA code, was convolved with the Instrument Spectral Response Function (ISRF) of the IASI-NG instrument. Subsequently, the Noise Equivalent Spectral Radiance (NESR) and the Absolute Radiometric Accuracy (ARA) were added.

- **Lines 204-213: This paragraph contains several unexplained abbreviations.**

We added the full description of the abbreviations and acronyms.

- **Line 211: Why 1.4 km? Please explain.**

We modified the text in order to clarify this point:

CAIRT instrument specifications have been defined based on performance optimization during the ESA study CAIRT-SciReC, in the frame of which also these simulations have been performed. As an outcome of this, the CAIRT instrument specifications have been simulated by application of a vertical SEDF (System Energy Distribution Function) as a Gaussion function of 1.4 km FWHM to the KOPRA pencil-beam simulations.

- **Line 223: Although well-explained, ALT is a somewhat confusing abbreviation for the along-track position, as it is often used for the vertical (altitude) dimension.**

We substituted ALT with AKT.

- **Line 342: Which layers?**

We made a mistake in breaking the paragraph in the text. We corrected it so that it is now clear that "layers" is referred to the altitude levels below 4 km.

- **Data availability: Possibly the reader can already be referred to the foreseen IASI-NG and CAIRT data archives, if already existing?**

We thank the Reviewer for the suggestion. The IASI-NG instrument was recently launched onboard the MetOp-SG satellite on 13 August 2025; however, its data are not yet publicly available as the mission is still in its commissioning phase. The CAIRT instrument is, instead, still a candidate mission for ESA's Earth Explorer

programme (Cycle 11) and is awaiting the outcome of the selection process. For these reasons, it is not yet possible to provide references to dedicated data archives in the current version of the manuscript.

---

## Author Comment (AC2)

**Reviewer 2**

\*\*\*\*\*\*\*\*\*\*\*\*\*\*\*\*\*\*\*\*\*\*\*\*\*\*\*\*\*\*\*\*\*\*\*\*\*\*\*\*\*\*\*\*\*\*\*\*\*\*\*\*\*\*\*\*\*\*\*\*\*\*\*\*\*\*\*\*\*\*\*\*\*\*\*\*\*\*\*\*\*\*\*\*\*\*\*\*

We thank the reviewer for the useful comments. In the following, we answer the specific comments (included in **"boldface"** for clarity) and, whenever required, we describe the related changes implemented in the revised manuscript. Page and line numbers indicated refer to the revised version of the paper. The text added in the revised version of the paper to address the reviewers' comments is highlighted in blue.

\*\*\*\*\*\*\*\*\*\*\*\*\*\*\*\*\*\*\*\*\*\*\*\*\*\*\*\*\*\*\*\*\*\*\*\*\*\*\*\*\*\*\*\*\*\*\*\*\*\*\*\*\*\*\*\*\*\*\*\*\*\*\*\*\*\*\*\*\*\*\*\*\*\*\*\*\*\*\*\*\*\*\*\*\*\*\*\*

General Comments

- **The description of the vectors in Section 2.3 could be support with a diagram to highlight the difference between the retrieval products from the limb measurements and from the nadir measurements. It would also be good in this section or earlier to specify what your retrieval targets are.**

We specified the retrieval target in the introduction and at line 127 (Sect. 2.3) and we added a graphical representation of the retrieval products of 1D and 2D tomographic retrieval (Fig.1) in accordance with the reviewer request.

- **Section 3.1 requires references for the instrument details. While all the relevant information is there, this section is hard to follow as the instruments are introduced and key information about them is then merged in the second paragraph. I would recommend that this section be separated as follows: introduction to why CAIRT (aims, details, advantages etc), introduction IASI-NG (aims, details advantages etc.), why the synergy is useful (similar to your current last paragraph).**

We rewrite the instruments section following the reviewer's request.

- **In Section 3.2 it is unclear why the ERS are being introduced, how many simulations you are performing, and what was performed in Ererra, 2023 vs in this work. It would be better to include an introduction paragraph for this section to describe the use of these simulations and to justify why two different radiative transfer models are being used. Figures of the ozone profiles should also be included.**

We simulated the IASI-NG and CAIRT for only one ERS scenario (line 207), we introduced the ERS datasets to describe how these scenarios are built. The citation Errera, 2023 refers to the published complete database, so that the reader could access directly to the data with the corresponding documentation.

The complete data fusion algorithm is an a posteriori method to combine Level 2 products supplied by the individual retrieval processors of the independent measurements. It is common that different Forward Models (FMs) are used in the independent retrievals. We describe and clarify this in the text. Different forward models (FMs) were used for practical reasons: limb and nadir observations were simulated by two separate groups within the project framework, each employing its own FM. However, this difference is not relevant to the objectives of the present study.

We added an introduction paragraph to Section 3.2:

To evaluate the performance of the extended CDF algorithm, we conducted a series of simulations based on synthetic measurements from the CAIRT and IASI-NG instruments. These simulations were necessary due to the unavailability of IASI-NG and CAIRT real data. The simulated datasets reproduce realistic observational scenarios and allow us to test the algorithm under controlled conditions. In the following, we describe the

simulation setup, including the forward models, instrument characteristics, and the atmospheric scenario used for generating the synthetic measurements. The CDF algorithm combines Level 2 products provided by the individual retrieval processors of the independent measurements. It is common for different Forward Models (FMs) to be used in the separate retrievals.

A new figure (Fig. 2) shows the ozone profile for the selected ERS scenario, along with the a priori profile.

- **Section 6: While the results look promising, it is difficult to interpret the improvements in DFS and the AKMs without information about the ozone profiles. It is not explained why the measurements in these three cases are suspected to be representative of the CAIRT and IASI-NG fusion. More detail is needed about the acquisitions themselves and why they have been averaged in this way. In this section you also refer to the IASI-NG and CAIRT measurements, please be specific that these are synthetic measurements or simulations.**

We added a figure with the true profile of the ERS scenario (Fig. 2), in order to better evaluate the improvements in the AKMs and DFS. The aim of this study is to describe the extension of the Complete Data Fusion (CDF) algorithm to two-dimensional (2D) retrieval products and to assess its performance, particularly in comparison with its application to one-dimensional (1D) products. To this end, we analyse the diagonal elements of the averaging kernel matrices and the total retrieval error, and we quantitatively evaluate the synergy between measurements using the Synergy Factors.

The three test cases presented are representative of the configurations described (in terms of objectives and specific measurements characteristics) in the three subsections of the Results section (from lines 289 to 313). To facilitate understanding, we have also included schematic figures illustrating the geometry of each case, as requested by Reviewer 1 (Fig. 3, 5, 8).

We modified the text in order to clarify in each sentence that we are using synthetic measurements.

########################################

Technical Comments

- **Line 21: It's not necessary to cite all of these references, only cite the key ones.**

The selected scientific articles report on studies that are particularly relevant to the investigation of synergies among different remote sensing measurements, each addressing different aspects of the synergy. If the reviewer has suggestions regarding which references should be considered as key, we are open to revising our selection accordingly. Otherwise, we would prefer to retain the current set of references.

- **Line 35: It is not initially clear what the features (that you explain in the next sentence) are. Please combine these two sentences and rewrite.**

We modified the sentence as suggested (lines 36-40):

The CDF is termed "complete" because it accounts for all features of the combined measurements, namely the retrieval errors of the fusing profiles and their correlations (represented by the covariance matrices, CMs) and the sensitivity of the retrieved profiles to the true profile (described by the averaging kernel matrices, AKMs), and can be regarded as a generalization of the weighted mean for cases where AKMs differ from the identity matrix.

- **Line 47: Can you define what Multi Target Retrieval is?**

We added the definition of Multi Target Retrieval in the text.

- **Line 90: Define Sn,I and Ss,i**

We modified the text accordingly to the reviewer comment.

- **Line 113: This looks like the same citation twice.**

The cited papers are different:

Ceccherini, S.: Comment on "Synergetic use of IASI profile and TROPOMI total-column level 2 methane retrieval products" by Schneider et al. (2022), Atmospheric Measurement Techniques, 15, 4407–4410, https://doi.org/10.5194/amt-15-4407-2022, 2022.

Ceccherini, S., Zoppetti, N., and Carli, B.: An improved formula for the complete data fusion, Atmospheric Measurement Techniques, 15, 7039–7048, https://doi.org/10.5194/amt-15-7039-2022, 2022.

- **Line 127: Equation 14 and the description above could be better explained. If my understanding is correction, this could be rewritten like the below.**

   **X^t = [pjk] where j=[1,m] and k=[1,n]**

The correction suggested by the reviewer does not fully reflect the structure of the state vector as defined in Equation 14, since it overlooks the precise ordering of parameters, which is rigorously specified by the original formulation. For this reason, we recommend retaining the original equation.

- **Line 137: I would suggest moving this to the supplementary or removing it. This does not seem particularly important for the paper.**

The description of the averaging kernel matrix (AKM) arrangement is crucial for a proper understanding of how the CDF algorithm is applied to two-dimensional products. A rigorous and detailed explanation is therefore essential to support the methodological focus of this paper.

- **Line 163: Change your definition of ranges to '…645 to 2760 cm1 (15.5 to 3.6 microns)' for easier reading**

We modified the text as requested.

- **Line 185: Please specify if KLIMA is a line-by-line or fast rt model here, what version of the model you are using, and what HITRAN database it is based on.**

We added the information about KLIMA as requested:

KLIMA is a line-by-line model and for these simulations we used the AER v3.8.1 spectroscopic database.

- **Line 220: It would be good to see a plot of the NESR and target ARA for this and IASI-NG in Section 3.1.**

We have added references to the NESR specifications for both CAIRT and IASI-NG in the Simulations Section. ARA is not defined for CAIRT. Instead, a Radiometric Scaling Error is defined, which is an error proportional to the radiance, and a Radiometric Additive Error, which accounts for error contributions that are independent of the radiance level. We added a reference to a Figure in Ridolfi et al. (2022) showing the ARA

of IASI-NG. We believe that a detailed discussion of the technical characteristics of the two instruments goes beyond the scope of this paper, which focuses on the application and evaluation of the CDF algorithm.

- **Line 277: This information should be presented more clearly in a table in Section 4.**

We added Table 1, with these information.

- **Figure 1/3: The light blue is hard to see and the blue is not clear on the figure. Where has your aprori error come from? What do the profiles look like? (lines 227 for the apriori description)**

We changed the cyan colour in the plots with green. We modified the caption of Figure 1, adding a reference to McPeters and Labow for the a priori error profile.

The blue line is not visible in Figure 1 as the AKM diagonal elements profile for the fused product is superimposed to that of CAIRT as written in line 325-327 and the total error profile of the fused product is overlapped to that of CAIRT (lines 333-334).

The true profile on which the analysis is performed is reported in figure 2.

- **Line 351: The description of a synergy factor should be moved to Section 5 where you introduced other quantifiers of performance.**

We moved the description of the synergy factor in accordance with the reviewer's request.

---

## Author Response (AR2)

Dear Editor,

in response to your request, we have expanded the captions of Figures 3, 5, and 8, and revised the captions of Tables 2, 3, and 4. We also confirm that the year of publication of McPeters and Labow is 2012 in both the text and the reference list.

Thank you for your attention.

Best regards,

Cecilia Tirelli